# Problem-solving therapy for pregnant women experiencing depressive symptoms and intimate partner violence: A randomised, controlled feasibility trial in rural Ethiopia

**Roxanne C. Keynejad**[1]*, **Tesera Bitew**[2,3], **Katherine Sorsdahl**[4], **Bronwyn Myers**[5,6,7], **Simone Honikman**[4,8], **Girmay Medhin**[9,10], **Negussie Deyessa**[2], **Adiyam Mulushoa**[9], **Eshcolewyine Fekadu**[9], **Louise M. Howard**[1], **Charlotte Hanlon**[2,9,11]

1 Health Service and Population Research Department, Institute of Psychiatry, Psychology & Neuroscience, Section of Women's Mental Health, King's College London, Denmark Hill, London, United Kingdom, 2 Department of Psychiatry, School of Medicine, College of Health Sciences, Addis Ababa University, Addis Ababa, Ethiopia, 3 Department of Psychology, College of Education and Behavioural Sciences, Injibara University, Injibara, Ethiopia, 4 Department of Psychiatry and Mental Health, Alan J Flisher Centre for Public Mental Health, University of Cape Town, Cape Town, South Africa, 5 Department of Psychiatry and Mental Health, Division of Addiction Psychiatry, University of Cape Town, Cape Town, South Africa, 6 Curtin enAble Institute, Curtin University, Bentley, Western Australia, Australia, 7 Alcohol, Tobacco and Other Drug Research Unit, South African Medical Research Council, Cape Town, South Africa, 8 Department of Psychiatry and Mental Health, Perinatal Mental Health Project, University of Cape Town, Cape Town, South Africa, 9 Centre for Innovative Drug Development and Therapeutic Trials for Africa (CDT-Africa), College of Health Sciences, Addis Ababa University, Addis Ababa, Ethiopia, 10 Aklilu Lemma Institute of Pathobiology, Addis Ababa University, Addis Ababa, Ethiopia, 11 Centre for Global Mental Health, Health Service and Population Research Department, Institute of Psychiatry, Psychology & Neuroscience, King's College London, Denmark Hill, London, United Kingdom

* roxanne.1.keynejad@kcl.ac.uk

**Data Availability Statement:** Anonymised data will be freely available on the King's Open Research

## Abstract

Evidence for the feasibility of brief psychological interventions for pregnant women experiencing intimate partner violence (IPV) in rural, low-income country settings is scarce. In rural Ethiopia, the prevalence of antenatal depressive symptoms and lifetime IPV are 29% and 61%, respectively. We aimed to assess the feasibility and related implementation outcomes of brief problem-solving therapy (PST) adapted for pregnant women experiencing IPV (PST-IPV) in rural Ethiopia, and of a randomised, controlled feasibility study design. We recruited 52 pregnant women experiencing depressive symptoms and past-year IPV from two antenatal care (ANC) services. Consenting women were randomised to PST-IPV (n = 25), 'standard' PST (not adapted for women experiencing IPV; n = 12) or enhanced usual care (information about sources of support; n = 15). Masked data collectors conducted outcome assessments nine weeks post-enrolment. Addis Ababa University (#032/19/CDT) and King's College London (#HR-18/19-9230) approved the study. Fidelity to randomisation was impeded by strong cultural norms about what constituted IPV. However, recruitment was feasible (recruitment rate: 1.5 per day; 37% of women screened were eligible). The intervention and trial were acceptable to women (4% declined initial screening, none declined to participate, and 76% attended all four sessions of either active intervention). PST-IPV was acceptable to ANC providers: none dropped out. Sessions lasting up to a

Data System at the following DOI: 10.18742/24047583.

**Funding:** This work was supported by a King's IoPPN Clinician Investigator Scholarship to RK. This work was supported by funding from the United Kingdom's National Institute for Health and Care Research (NIHR) to LH, the NIHR Global Health Research Unit on Health System Strengthening in Sub-Saharan Africa (ASSET), King's College London (GHRU 16/136/54) to RK and CH, the NIHR Global Health Research Group on Homelessness and Mental Health in Africa (NIHR134325) and the SPARK project (NIHR200842) both to CH using UK aid from the UK Government. This work was supported by funding from Debre Markos University and Injibara University to TB. This work was supported by the DELTAS Africa Initiative [DEL-15-01] to TB and KS, an independent funding scheme of the African Academy of Sciences' Alliance for Accelerating Excellence in Science in Africa, with support from the New Partnership for Africa's Development Planning and Coordinating Agency with funding from the Wellcome Trust and UK government. This work was supported by joint research grant funding from the Medical Research Council (MRC), Wellcome Trust NIHR and United Kingdom's Department for International Development (MR/M014290/1; MR/R018464/1) to KS and BM. This work was supported by funding from Addis Ababa University (AAU) to GM. This work was supported by funding from philanthropic organisations, Goldsmiths University, UKRI (EP/T030429/1), World Childhood Foundation, Bill and Melinda Gates Foundation and UKAID to SH. This work was supported by funding from UKRI to LH. This work was supported by grants from the Wellcome Trust (222154/Z20/Z and 223615/Z/21/Z) to CH. The funders had no role in study design, data collection and analysis, decision to publish, or preparation of the manuscript. The views expressed in this publication are those of the authors and not necessarily those of the funders.

**Competing interests:** The authors have declared that no competing interests exist.

mean 52 minutes raised questions about the appropriateness of the model to this context. Competence assessments recommended supplementary communication skills training. Fidelity assessments indicated high adherence, quality, and responsiveness but assessing risks and social networks, and discussing confidentiality needed improvement. Adjustments to optimise a future, fully powered, randomised controlled trial include staggering recruitment in line with therapist availability, more training on the types of IPV and how to discuss them, automating randomisation, a supervision cascade model, and conducting post-intervention outcome assessments immediately and three months postpartum. Registration: Pan African Clinical Trials Registry #PACTR202002513482084 (13/12/2019): https://pactr.samrc.ac.za/TrialDisplay.aspx?TrialID=9601.

# Introduction

## Perinatal common mental disorders

Common mental disorders (CMDs): depression, anxiety disorders, and post-traumatic stress disorder (PTSD) are common during pregnancy and associated with maternal, foetal, and neonatal morbidity and mortality [1]. A meta-analysis of 28 studies of antenatal depression in ten African countries (n = 17,938 women) found a pooled prevalence of 26.3% [95% confidence interval (CI) = 22.2–30.4; 2]. Antenatal depression was associated with 'unfavourable marital condition' (p. 8), including marital conflict (odds ratio (OR) = 4.17, CI = 1.75–9.94). These prevalence estimates for CMDs were higher than the 10% reported antenatally [3] and 13% postnatally in high-income countries [HICs; 4]. In rural Ethiopia, a population-based prospective study found that 28.7% of pregnant women screened positive for antenatal depressive symptoms [5]. Incident depression was associated with intimate partner violence (IPV) during pregnancy (adjusted risk ratio (aRR) = 1.06, CI = 1.00–1.12 [5]).

## Perinatal intimate partner violence

IPV causes physical, sexual and psychological harm, and affects around 30% of ever-partnered women worldwide [6]. A global meta-analysis (92 studies) found average reported IPV prevalence of 28% (emotional abuse), 14% (physical abuse) and 8% (sexual abuse) during pregnancy [7]. A second meta-analysis (67 studies) found increased odds (OR = 3.1) of postpartum depressive symptoms in women experiencing IPV during pregnancy and higher odds of experiencing IPV in women experiencing perinatal CMDs [8]. In Ethiopia, the combination of depression and maternal IPV exposure is associated with higher child mortality (relative risk (RR) = 4.0 for physical IPV and 3.7 for emotional IPV) than depression alone [RR = 2.3; 9]. Given direct mortality from suicide and complications of substance use, indirect mortality from obstetric complications [10,11], and increased child mortality, the mental health of pregnant women experiencing IPV is a priority.

## Brief psychological interventions

Brief psychological interventions are condensed, simplified versions of talking therapies, which were often originally developed to be delivered over several months. Given healthcare resource constraints and very limited availability of mental health specialists in LMICs, the World Health Organization's mental health gap action programme intervention guide (mhGAP-IG) recommends a range of evidence-based brief psychological interventions [12].

An umbrella review of meta-analyses of studies conducted in low and middle-income countries (LMICs) identified 123 primary randomised controlled trials [RCTs; 13]. Highly suggestive evidence of moderate credibility supported the efficacy of such brief psychosocial interventions in adults with CMDs in general settings (standardised mean difference = 0.49, CI = 0.36–0.62).

**Trauma-informed care.**   Generic psychological interventions can neglect the role played by trauma in the mental ill-health of women experiencing IPV. Traumatic experiences (such as IPV) have lasting physical and psychological impacts, which influence a person's relations to the world, self and others [14]. People with lived experience of IPV have cautioned against 'pathologising' IPV-related difficulties, by considering mental health conditions in isolation, without addressing the emotional impact of IPV [15]. Neglecting the wider needs of people experiencing IPV may reduce psychological interventions' therapeutic potential [16]. Trauma-informed care prioritises trauma awareness, safety, trustworthiness, choice, collaboration, strength, and skill-building [17].

Very few RCTs have evaluated psychological interventions for CMDs, tailored for women's experiences of IPV in LMICs. Studies from Tanzania [18], Pakistan [19,20], and Iran [21] support their potential benefits, but only one pilot RCT from Nepal focused specifically on pregnant women [22]. We therefore conducted qualitative research [23], identifying a South African model of brief problem-solving therapy [PST; 24] as most contextually relevant to rural Ethiopia, and followed UK Medical Research Council (MRC)-affiliated 'ADAPT' guidance [25] to adapt it for the needs of pregnant women experiencing IPV in this setting.

**Integrating mental health into antenatal care.**   In low-resource settings, antenatal care (ANC) may be women's only contact with healthcare [26]. Regular appointments, continuity of care providers and postpartum follow-up can facilitate building trust and rapport [27], which are essential for women to discuss sensitive subjects. ANC therefore offers an important opportunity for intervention. However, most perinatal psychological interventions for women experiencing IPV come from high-income countries [26]. Although some have been evaluated in LMICs l [28,29], few use RCT methods. A systematic review of interventions for pregnant women reporting IPV in LMICs [30] identified only one (quasi-experimental) study which measured depressive symptoms [31]. A meta-analysis of RCTs of psychological interventions which measured IPV exposure in LMICs [32] found only three studies enrolling pregnant women; pregnancy was sometimes an exclusion criterion. Brief psychological interventions tailored for pregnant women experiencing IPV in LMICs therefore require development and evaluation.

**Feasibility studies.**   The latest UK MRC and National Institute for Health and Care Research (NIHR) guidance on developing and evaluating complex interventions [33] advocates careful intervention development or identification, followed by feasibility assessment, before definitive evaluation, and finally, implementation. 'Core elements' required in each of these four phases include considering the context, articulating the intervention's programme theory, engaging stakeholders, identifying uncertainties, and refining the intervention.

The need to determine the feasibility of research methods [33,34] and explore other implementation outcomes [35,36], before conducting fully-powered RCTs is increasingly recognised. A recent study piloted a health system strengthening intervention to improve antenatal care for pregnant women experiencing CMDs and IPV in South Africa: an upper middle-income country [37]. The authors found that although the intervention was acceptable and appropriate, the feasibility, adoption, and fidelity of delivery were poor. In particular, task-sharing was not a feasible model to integrate mental healthcare for women experiencing IPV into ANC; adoption and fidelity were impeded by clinicians already struggling to fulfil their existing workloads. Evidence from low-income countries regarding the feasibility of perinatal mental health interventions for women experiencing IPV remains scarce.

## Objectives

We aimed to assess the feasibility and related implementation outcomes of brief problem-solving therapy (PST) adapted for pregnant women experiencing IPV (PST-IPV) in rural Ethiopia, and of a randomised, controlled feasibility study design. In previous work, we considered the Ethiopian context and the most appropriate intervention model through formative qualitative research, stakeholder engagement, and a theory of change [38]. We identified a model of brief PST which had been effective in a range of South African settings [24,39] as best suited to this context. We previously followed MRC-affiliated 'ADAPT' guidance [25] to adapt brief PST for the needs of pregnant women experiencing IPV in rural Ethiopia. In the present study, we hypothesised that:

1. The PST-IPV intervention and the processes of participant recruitment, randomisation, and follow-up would be feasible to deliver in rural Ethiopia.

2. The intervention and randomised, controlled design would be acceptable to pregnant women, with a drop-out rate of 30% or less.

3. The intervention and trial design would be acceptable to ANC providers, appropriate to the setting, and be delivered safely, with fidelity to the intended model and trial standard operating procedures.

## Material and methods

### Trial design

We conducted a three-arm randomised, controlled feasibility trial [40], the processes of which are summarised in Fig 1. This study was linked to a two-arm feasibility trial [41], which randomised women screening positive for depressive symptoms and functional impact to 'standard' PST (not adapted for women experiencing IPV) or enhanced usual care (EUC).

### Participants

**Setting.**   We conducted this study in Sodo district, in the Southern Nations, Nationalities and People's Region (SNNPR) of Ethiopia. ANC is delivered in Ethiopia through a network of satellite 'health posts', health centres, and primary hospitals. ANC providers in health centres comprise midwives, health officers (clinicians with four years' training), and nurses.

**Eligibility criteria.**   Participating women were required to speak Amharic, be aged 16 years or older, be between 12 and 34 weeks pregnant, and intending to live in Sodo for the duration of the study. Women were eligible to participate if they scored five or more on the locally adapted [42] Patient Health Questionnaire [PHQ-9; 43] and endorsed a tenth item regarding the functional impact of symptoms. This score was the optimal cut-off for identifying possible depressive disorder in a validation study conducted in Sodo [44].

To be eligible, women were also required to report IPV in the past year (endorsing any item on a five-item 'non-graphic language' screen, previously found to be a valid measure of IPV in this and other LMIC settings [45], or any item from the World Health Organization (WHO) multi-country study of women's health and domestic violence instrument, previously used in this setting [46]. An example item on the non-graphic language screen is "how is your partner treating you (and your children)?" (Always well/well most of the time/neutral/not well most of the time/never well). An example item on the WHO instrument is "the next questions are about things that happen to many women, and that your husband may have done to you. I want you to tell me if your husband or any other partner has ever done the following things to

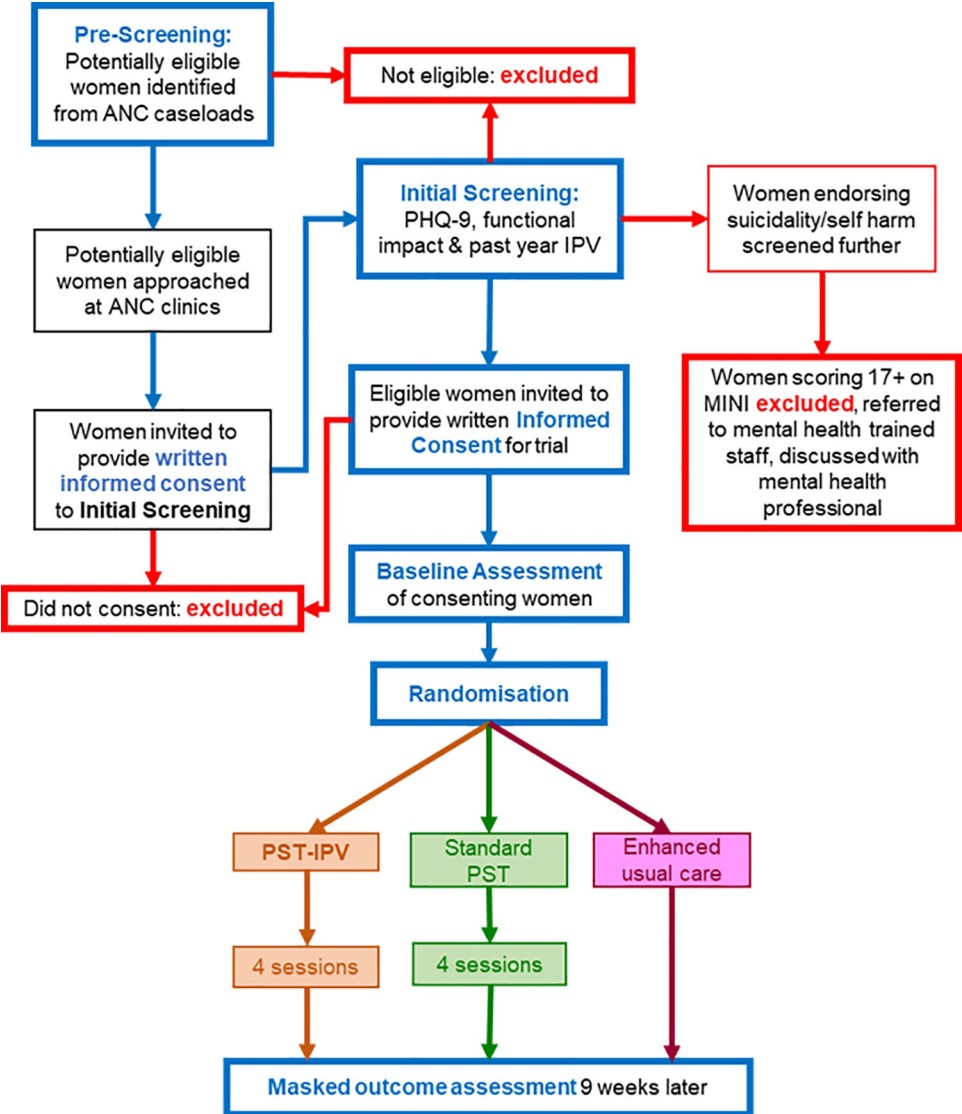

**Fig 1. Trial flow diagram.** ANC: Antenatal care, PHQ-9: Patient health questionnaire, MINI: Mini international neuropsychiatric interview, IPV: Intimate partner violence, PST: Problem-solving therapy, PST-IPV: PST adapted for women experiencing IPV.

you in the past 12 months. (a) Insulted you or made you feel bad about yourself? (Yes/No. If yes, how many times in the past 12 months)?"

Women were excluded from participation if they were acutely, physically ill, required emergency treatment of any kind, were identified by their ANC provider during pre-screening as displaying possible psychotic symptoms, could not understand the information provided (for example, due to intellectual disability, or inability to speak Amharic) or did not expect to remain in the study area for the nine weeks following recruitment. Any women assessed as being at imminent risk from physical or mental ill-health were referred to the primary hospital, for escalation of their care.

**Recruitment and consent.** Data collectors pre-screened the clinical records of pregnant women treated by ANC providers at two sites (Bu'i primary hospital and Kela health centre) in rural Sodo district, supervised by field supervisors. Potentially eligible women were then

invited to provide written, informed consent to receive initial screening for eligibility. Provision was made for non-literate women to consent by reading information aloud and independently witnessed thumbprinting.

Women who endorsed self-harm or suicidal ideation (PHQ-9 question nine) were screened further using the Mini International Neuropsychiatric Inventory [MINI; 47]. Women scoring 17 or more, indicating a high suicide risk, were excluded from participation [48]. They were referred to a member of primary care staff trained to use WHO mhGAP-IG criteria [12] to assess for imminent risk of self-harm or suicide. Women assessed as being at imminent risk were referred to a psychiatric nurse at the primary hospital, for escalation of their care. Any psychotic symptoms and other risks identified during the trial were discussed with the project mental health professional and women were referred for specialist mental healthcare, if required. All research staff followed a standard operating procedure (SOP), including protocols for responding to IPV risks.

Women identified by initial screening as eligible to take part in the trial were required to provide written, informed consent after being provided with study information by data collectors. As with initial screening, non-literate women indicated their consent by independently witnessed thumbprinting, after study information had been read aloud.

## Interventions

All arms represented additions to standard clinical care, where no routine interventions were provided for perinatal depression or IPV. EUC comprised a leaflet on well-being during pregnancy and information about free, local sources of support available to women and families, provided by the field supervisor at the time of randomisation. As no specific perinatal mental health or IPV services were available at the time of the study, sources of support constituted general provision such as the government-funded Women and Child Affairs Office.

**Active arms.**   Participants randomised to PST-IPV or standard PST were offered four intervention sessions delivered by trained, supervised ANC providers, who were paid for their time. All four sessions were delivered within eight weeks from the date of enrolment. The process by which we adapted PST-IPV [23] and standard PST [49] are described in detail elsewhere. Both active arms were four-session manualised interventions based on a model successfully implemented in South Africa [24]. During sessions, women were guided to identify the most important things in their lives, list and categorise their problems, learn and practise coping strategies for different problem types. The content of each of the four PST-IPV sessions is described in S1 Table.

PST-IPV differed from standard PST in that ANC providers attended a trauma and IPV-informed training course, received education about IPV in the intervention manual, learned how to respond to IPV-related problems and to avoid inadvertently colluding with abuse or victim blaming during intervention sessions, and paid attention to IPV and risks, including during supervision.

**Training, competence assessment, and supervision.**   Six ANC providers received five days' PST-IPV training, delivered by a local psychiatrist and psychologist with experience of delivering mental health training to primary care staff. To limit contamination between arms, a separate group of six ANC providers received standard PST training, delivered by a different psychiatrist and psychologist to PST-IPV, using the same five day structure.

Both PST-IPV and standard PST training courses included sessions guiding ANC providers through paper copies of the intervention manual and a desktop flip-chart resource (with contextually relevant illustrations facing the woman and written instructions facing the ANC provider). Small group discussions, activities, and role play tasks incorporating peer feedback were used during training.

With participants' agreement, ANC providers audio-recorded each session of PST-IPV or standard PST. Anonymised session recordings were then uploaded to the study computer and securely shared with their supervisor (the psychiatrist or psychologist who had trained them), to inform the content of supervision sessions. After the training course, PST-IPV and standard PST-trained ANC providers needed to be approved by their supervisor as competent, based on audio recordings of them delivering an 'accelerated case' of four sessions within a two-week period.

After completing a first, accelerated, case of PST-IPV or standard PST, and being assessed as competent to continue delivering sessions, ANC providers received regular supervision. The lead researcher (RK) remained in weekly email or videoconferencing contact with all four supervisors, to monitor the progress of supervision and discuss any concerns.

A sample of anonymised session audio recordings was evaluated by two trained independent psychologists not otherwise involved in the study, to examine competence in non-specific therapeutic skills, using relevant items from the Enhancing Assessment Of Common Therapeutic Factors (ENACT) rating scale [50], adapted for Ethiopia [51]. Audio recordings were also evaluated for session fidelity and completion, using a checklist of expected session components; the frequency and duration of IPV-specific content was noted.

## Outcome measurement

Assessments were completed at baseline and nine weeks later, in a private room of the healthcare facility. Outcome assessments were performed by independent data collectors blinded to participant allocation. Data transferred to the study authors used anonymised participant identifiers and no personally identifying information, ensuring that participants could not be personally identified.

**Demographic variables.** Data collectors recorded demographic information obtained from a list kept by HEWs. At baseline, we recorded participants' gestation, gestation at the time of their first ANC visit, the proportion of recommended ANC appointments they had attended, any physical or mental health medications prescribed, any documented past psychiatric history or current mental health conditions, and any known obstetric complications. Post-partum, data collectors obtained information regarding any obstetric or neonatal complications affecting participants.

Additional demographic data obtained by asking participants directly comprised women's age, religious affiliation, district of residence, their own and their partner's highest education level, their own and their partner's occupation, marital status, age at marriage, gravidity, parity, whether their pregnancy was planned, (if planned) whether the pregnancy had occurred at their preferred time, and total occupants in their household. Data collectors also administered the Amharic-translated list of threatening experiences [52] at baseline, to capture past-year exposure to stressors other than IPV.

**Feasibility and other implementation outcomes.** In this feasibility study, the primary outcome was the overall feasibility of implementing PST-IPV and the randomised, controlled study design. Secondary outcomes were the acceptability, appropriateness, fidelity, and safety of PST-IPV and the trial design. We operationalised each of these outcomes as follows, informed by the definitions in Proctor and colleagues' taxonomy [36]:

- Feasibility: recruitment rate (feasibility of recruiting the required sample size), outcome assessment completion, supervision attendance.

- Acceptability: uptake of initial screening, proportion of eligible women participating, comprehension and acceptance of randomisation (recorded in field journals by research staff

screening women and inviting them to participate), ANC provider drop-out rate, proportion completing four sessions, acceptance of audio-recording.

- Appropriateness: recruitment rate (numbers of eligible women identified who also consented to take part), mean session duration, frequency and duration of IPV-specific content.

- Fidelity: independent ratings of fidelity and competence, independent assessors' qualitative feedback, supervisors' qualitative feedback.

- Safety: frequency of physical health, mental health, and safety-related adverse events. As previous research in this setting found that perinatal complications were not reliably self-reported [53], births, stillbirths, and early neonatal deaths were collected from HEWs, who maintain maternal care records. Any mental health and safety concerns identified during the study were discussed with the project mental health professional, who kept a log and followed up adverse events.

- We also noted the feasibility of implementing the study design during the coronavirus pandemic and the acceptability of attending four intervention sessions in accordance with government requirements for mask wearing, hand washing, and physical distancing [54].

**Outcome measures: Feasibility and acceptability.** In addition to our primary focus on feasibility and implementation outcomes, we also piloted the use of instruments at baseline and outcome assessments, to determine their feasibility and acceptability, noting any concerns about respondent burden raised by data collectors or participants. All instruments were translated into Amharic.

We measured depressive symptoms with the PHQ-9 [43,44], and post-traumatic stress symptoms using the PTSD Checklist for DSM-5 [PCL-5; 55], which has been adapted for a rural Ethiopian context. We measured anxiety symptoms using the Generalised Anxiety Disorder Scale [GAD-7; 56] and functional impact using the 12 item Ethiopian adaptation [57] of the WHO Disability Assessment Schedule [WHODAS 2.0; 58].

**Pilot testing of hypothesised mediators.** Through our formative work informing our adaptation of the intervention and articulating its 'theory of change,' we identified several hypothesised mediators, by which PST-IPV might improve perinatal depressive symptoms in women experiencing IPV in this context [23]. We therefore piloted the use of instruments measuring these constructs in baseline and outcome assessments.

We measured gender attitudes using the WHO Attitudes Towards Gender Roles Questionnaire [46]. We used an adapted scale validated in Ethiopia for sexual health research [59] to measure self-efficacy and the OSSS-3 Oslo scale [60], previously used in a study of antenatal depressive symptoms in Sodo [61], to measure perceived social support. We measured mastery (the extent to which the participant considered herself in control of forces affecting her life [62]) using a 15 item multicultural mastery scale [63] adapted for rural, non-Western communities.

Finally, we measured healthcare-seeking behaviour (including from non-biomedical sources such as religious healers, herbalists, and holy water sites), using the Client Service Receipt Inventory (CSRI) [64], adapted for Ethiopia [65] and modified to focus on the past three months, except for inpatient care at baseline, which was asked for the past 12 months.

## Sample size

As a feasibility trial, we did not seek to evaluate intervention efficacy, but to estimate feasibility parameters to inform a future RCT and test intervention and research protocols. We used

NIHR guidance [66] to calculate the percentage confidence interval (CI) with which our study could estimate the actual drop-out rate from a future RCT, as follows. CI = 1.96 x $\sqrt{}$ (p (1—p) / n), where p equalled the anticipated proportion of participants dropping out of the study and n was the intended sample size. Using this calculation, we determined that a total sample size of 75 across the entire trial, including the nested study (25 participants randomised to PST-IPV, 25 to standard PST and 25 to EUC) would enable us to estimate a drop-out rate of 30%, within a 95% CI of +/- 10%. That is, 0.1037 = 1.96 x $\sqrt{}$ (0.3 x 0.7 / 75). This sample size aligned with recommendations that sample size calculations for a future RCT require pilot trials with 12 to 25 participants per arm [67,68].

## Randomisation

**Sequence generation.**    A three-arm randomisation table was generated by the trial statistician (who was not otherwise involved in data collection processes) using a random number list. The table proportionately allocated women eligible for this trial (experiencing depressive symptoms and IPV) in the ratio of 2:1:1 to PST-IPV, standard PST, and EUC arms.

The separate, nested feasibility trial used a different (two-arm) randomisation table to allocate women experiencing depressive symptoms but not reporting IPV in a 1:1 ratio, to standard PST and EUC arms.

Women eligible to take part in either of the two trials were randomised using these tables until 25 women had been randomised to an arm, at which point enrolment into that arm was closed and recruitment continued for the remaining, unfilled arms.

**Allocation concealment.**    Randomisation tables were retained by the trial coordinator and research assistant, who were telephoned by field supervisors following each baseline assessment, to anonymously allocate that participant to an arm. Field supervisors were unmasked and followed the appropriate procedure, depending on which arm was allocated. They provided participants randomised to active arms with an unmarked, coloured card, indicating their enrolment in a trial arm. The field supervisor informed participants of the date and time at which their first session of PST-IPV or standard PST would take place, after agreeing this with their allocated ANC provider. Women allocated EUC received an appointment nine weeks later (to conduct an outcome assessment), on the day of randomisation.

**Masking.**    Given differences between PST-IPV, standard PST, and EUC arms, ANC providers delivering intervention sessions could not be masked to participant allocation. To avoid contamination, PST-IPV and standard PST were delivered by two different groups of ANC providers, who had attended either a PST-IPV or a standard PST training course.

Data collectors who had recruited participants at Bu'i primary hospital conducted post-intervention outcome assessments of women recruited at Kela health centre, and vice versa. Data collectors were masked to intervention allocation at the time of outcome assessment and documented any incidents of un-masking. Data analysts were also masked to intervention allocation.

**Reimbursement.**    Participants were reimbursed for their time to attend baseline and outcome assessments, and for any transport costs incurred by participating in the study. All payment fees were set by the Addis Ababa University (AAU) CDT-Africa finance department, aligned with government compensation scales.

## Analysis

As a feasibility study, this trial was not powered to calculate effect sizes. However, we calculated recruitment and drop-out rates, and appropriate measures of the central tendency of outcome measures for the intention-to-treat sample (all participants who were randomised,

regardless of their uptake of intervention sessions), using STATA [69]. We determined whether to calculate means and standard deviations, or medians and interquartile ranges, based on visual inspection of histograms for evidence of normal distribution or skew.

### Ethics and risk

King's College London (#HR-18/19-9230) and AAU (#032/19/CDT) institutional review boards (IRBs) provided ethical approval.

Our trial recruited potentially vulnerable women and entailed IPV disclosure and mental health symptom detection. The SOP (available from our open access repository [70]) therefore outlined systems to mitigate potential risks to participants. ANC providers delivering intervention sessions had access to a project mental health professional, with whom they could raise any immediate clinical concerns.

Research staff were trained to listen non-judgementally, offer privacy, confidentiality and information about sources of support to women disclosing IPV, following WHO guidelines [71]. The SOP outlined measures to mitigate risks to participants, including abusive partners learning about their involvement. These included a protocol for responding to concerns or disclosures of risk, emergency contact information, conduct when communicating with vulnerable participants, documenting and responding to serious adverse events.

A fund was available to assist participants disclosing IPV, severe mental health symptoms, or suicidal ideation to access support services, such as transportation to a government social support office, police station, or secondary mental health service.

### Registration

We registered the trial on 13/12/2019 (Pan African Clinical Trials Registry #PACTR202002513482084) and published the trial protocol on 01/06/2020 [40].

## Results

### Participant flow and recruitment

Recruitment took place over 35 days from April to May 2021, as summarised in Fig 2 (CONSORT diagram). S1 Fig (extended CONSORT diagram) summarises the flow of all participants in detail. S1 Table presents the CONSORT checklist.

Out of 335 pregnant women pre-screened, 193 (58%) were excluded for not meeting eligibility criteria, including 12 (4%) who declined initial screening. Of the remaining 142 women, a further 67 (47%) were excluded following initial screening. Of the remaining 75 pregnant women, 52 (69%) met inclusion criteria for our trial and were randomised to either PST-IPV, standard PST or EUC.

### Protocol deviations

In a deviation from the trial protocol, 17 out of the 52 women meeting inclusion criteria were misclassified as not experiencing IPV. For these participants, research staff mistakenly used the two-arm randomisation table reserved for women not reporting past-year IPV exposure (in the linked trial), instead of the three-arm table. This occurred because, despite clear protocols and training, some research staff interpreted IPV exposure to mean physical and/or sexual IPV only, and not psychological or emotional IPV. Therefore, although these 17 participants were randomised into one of two appropriate arms (standard PST or EUC), they were denied the opportunity of being randomised into the third trial arm (PST-IPV), as planned.

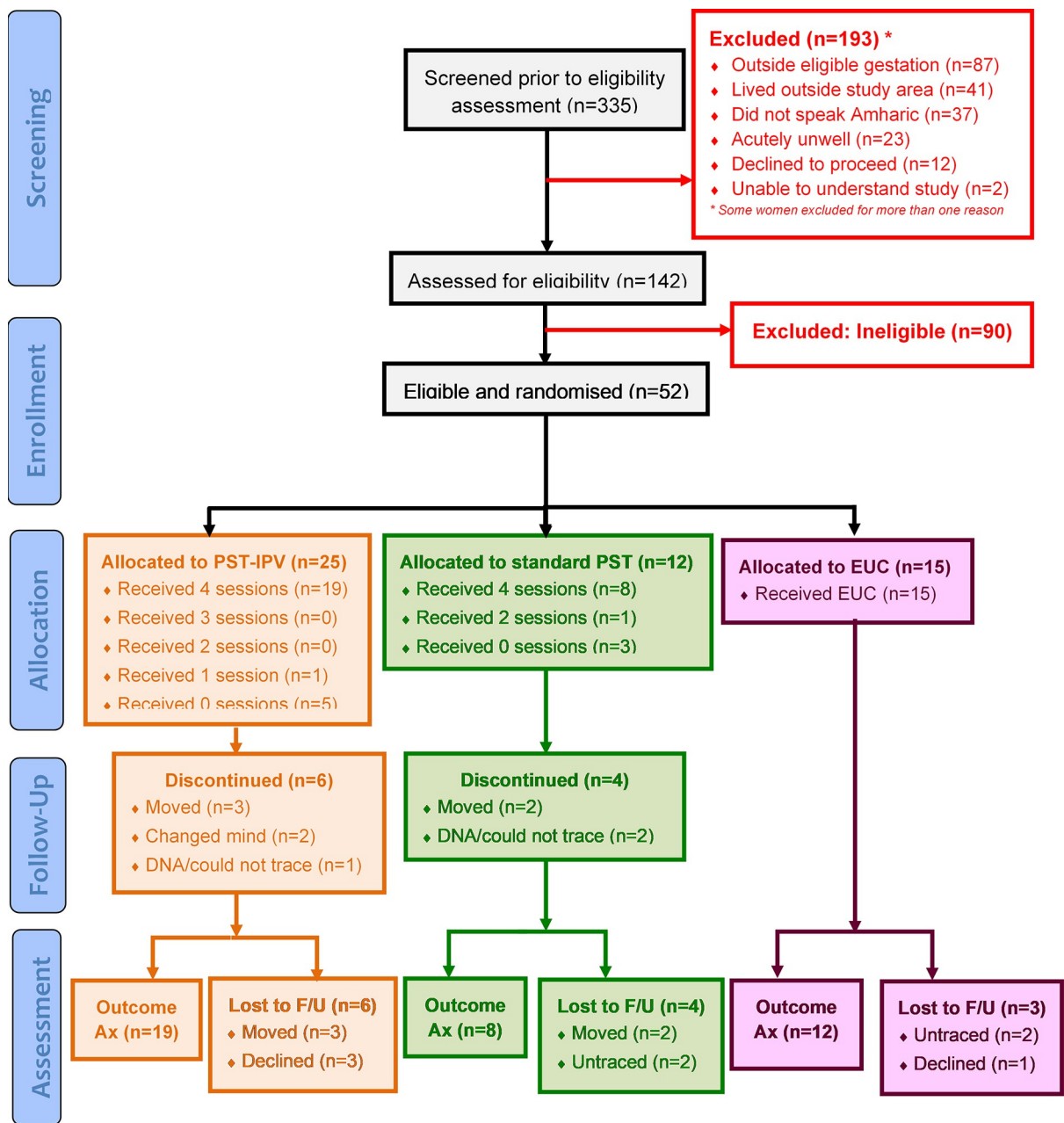

**Fig 2. CONSORT diagram.** Ax: Assessment, DNA: Did not attend, F/U: Follow-up, PST: Proble.m-solving therapy, PST-IPV: PST adapted for women experiencing IPV.

In a second protocol deviation, 12 out of 39 outcome assessments were conducted before nine weeks (between 37 and 61 days, rather than the target 63 days) had elapsed since the participant's enrolment in the trial.

## Participant characteristics

Table 1 shows demographic characteristics of 51 out of 52 participants, for whom data were available. The median age of participants ranged from 25 to 25.5 years across all study arms.

Most participants were Orthodox Christian, primary school educated, and gave their occupation as home-maker. Ninety-two percent of participants (n = 47) were in monogamous marriages and 72% were married by 20 years of age (n = 36).

Fifty percent of participants' partners worked as farmers (n = 25) and a further 26% as government or private sector employees (n = 13). Fifty-two percent of partners were educated to primary school level (n = 26), and median household members ranged from three to four persons across all arms.

Median gravidity ranged from two to 2.5 pregnancies, and median parity was one previous delivery across all arms. Fifty-seven percent of participants reported that their pregnancy was planned (n = 29); of these, 86% of participants reported that the timing of their pregnancy was also planned (n = 25). The median list of threatening experiences score was 22–22.5 across all arms (indicating between one and two threatening experiences).

No participant's medical records mentioned past or current mental health conditions, or IPV. Although the only documented medical comorbidity was dyspepsia (n = 1), 12 (23%) were prescribed physical health medications. No participant was prescribed psychiatric medication.

## Implementation outcomes

**Feasibility.** The feasibility of recruitment was demonstrated by enrolling 52 eligible participants over 35 days (1.5 participants per day). This required pre-screening 335 women and initially screening 142 women: recruitment rates per set of records pre-screened of 16% and per woman initially screened of 37%.

Outcome assessments were completed by 19 of 25 participants randomised to PST-IPV (76%), eight of 12 participants randomised to standard PST (67%), and 12 of 15 participants randomised to EUC (80%). Of the 13 participants out of 52 (25%) not completing outcome assessments, five (38%) moved away, four (31%) could not be traced, and four declined, usually due to a cultural practice of postnatal confinement. The median age of women who did not complete outcome assessments (n = 13) was 20 years (IQR = 18–26) in comparison to 25.5 years (IQR = 24–30) for those (n = 39) who did. Baseline IPV scores were similar (median NGL score was 4 in both groups, median WHO IPV score was 5 (IQR = 1–11) in the follow-up non-completion group and 6 (IQR = 3–10) in the completion group). Due to the small number of participants who did not complete the outcome assessment, we did not perform statistical comparisons. However, baseline trauma symptoms may have been higher in the group not completing outcome assessment (median PCL-5 = 9, IQR = 6–13) than in the group which did (PCL-5 = 6, IQR = 3–12). Baseline differences in anxiety (median GAD-7 = 11 (IQR = 9–12) compared to 10 (IQR = 7–12)) and depressive symptoms (median PHQ-9 = 9 (IQR = 8–10) compared to 8 (IQR = 7–11)) were less pronounced.

In terms of the coronavirus pandemic, the national state of emergency declared between April and October 2020, which limited the movement of people between geographical regions, delayed the initiation of the trial. Continuing restrictions sometimes prohibited in-person clinical supervision, leading to supervisors telephoning ANC providers instead of travelling to Sodo from Addis Ababa.

**Acceptability.** Of 335 women pre-screened prior to eligibility assessment, 12 (4%) who were potentially eligible to participate declined initial screening (see Fig 2). Of 142 women initially screened, none who were invited to participate in the trial declined. Participants comprehended and accepted randomisation, including to EUC.

PST-IPV was acceptable to pregnant women experiencing depressive symptoms and IPV: 19 out of 25 (76%) attended all four sessions. Reasons for attending one (n = 1) or no sessions

**Table 1. Demographic characteristics of participants at baseline.**

| Characteristic | PST-IPV (n = 25) | Standard PST (n = 12) | EUC (n = 15) |
|---|---|---|---|
| **Age in years** | (n = 24) | | |
| Median, IQR | 25.5 (23–28.5) | 25.5 (20–30) | 25 (21–29) |
| **Religion (n, %)** | (n = 24) | (n = 11) | |
| Orthodox Christian | 22 (92) | 9 (75) | 11 (73) |
| Protestant | 1 (4) | 3 (25) | 3 (20) |
| Muslim | 1 (4) | | 1 (7) |
| **Highest education (n, %)** | (n = 24) | | |
| Non-literate | 3 (13) | 2 (16.7) | 2 (13.3) |
| No formal education | 4 (17) | 1 (8.3) | 2 (13.3) |
| Primary school | 9 (38) | 8 (67) | 6 (40) |
| Secondary school | 6 (25) | 1 (8.3) | 3 (20) |
| Higher education | 2 (8) | | 2 (13.3) |
| **Occupation (n, %)** | (n = 24) | | |
| Homemaker | 18 (75) | 8 (66.7) | 6 (40) |
| Farmer | 1 (4) | 1 (8.3) | 2 (13) |
| Daily labourer | 2 (9) | 1 (8.3) | 1 (7) |
| Merchant/market seller | 1 (4) | 1 (8.3) | 2 (13) |
| Other | 2 (8) | 1 (8.3) | 4 (27) |
| **Marital status (n, %)** | (n = 24) | | |
| Married (monogamous) | 21 (88) | 12 (100) | 14 (93) |
| Married (polygamous) | 2 (8) | | 1 (7) |
| Cohabiting | | | |
| Single | 1 (4) | | |
| **Age at marriage (n, %)** | (n = 23) | | |
| 14 years old or younger | 1 (4) | | 1 (6.7) |
| 15–16 years old | 4 (17) | 2 (16.7) | 4 (26.7) |
| 17–18 years old | 8 (35) | 2 (16.7) | 1 (6.7) |
| 19–20 years old | 5 (22) | 5 (41.7) | 3 (20) |
| 21–25 years old | 5 (22) | 1 (8.3) | 6 (40) |
| 26–29 years old | | 2 (16.7) | |
| **Partner occupation (n, %)** | (n = 23) | | |
| Farmer | 12 (52) | 5 (41.7) | 8 (53.3) |
| Daily labourer | 5 (22) | | 1 (6.7) |
| Merchant/market seller | | 1 (8.3) | |
| Government employee | 6 (26) | 3 (25) | 4 (26.7) |
| Other | | 3 (25) | 2 (13.3 |
| **Partner's highest education level attended (n, %)** | (n = 23) | | |
| Non-literate | | | |
| No formal education | 3 (13) | 3 (25) | 2 (13.3) |
| Primary school | 13 (57) | 5 (41.7) | 8 (53.3) |
| Secondary school | 6 (26) | 3 (25) | 2 (13.3) |
| Higher education | 1 (4) | 1 (8.3) | 3 (20) |
| **Total household family members** | (n = 24) | | |
| Median (IQR) | 3 (3–5) | 3.5 (2.5–6.5) | 4 (2–6) |
| **Gestation in weeks at first ANC attendance** | (n = 25) | | |
| Median (IQR) | 20 (20–28) | 20 (18–28) | 26 (20–28) |
| **Gravidity** | (n = 24) | | |

*(Continued)*

**Table 1.**  (Continued)

| Characteristic | PST-IPV (n = 25) | Standard PST (n = 12) | EUC (n = 15) |
|---|---|---|---|
| Median (IQR) | 2.5 (2–4) | 2.5 (1.5–4) | 2 (1–3) |
| Range | 1–6 | 1–6 | 1–7 |
| **Parity** | (n = 24) | | |
| Median (IQR) | 1 (1–3) | 1 (0–1.5) | 1 (0–2) |
| Range | 1–5 | 0–5 | 0–6 |
| **Pregnancy planned (n, %)** | (n = 24) | | |
| Yes | 11 (46) | 8 (66.7) | 10 (66.7) |
| No | 13 (54) | 4 (33.3) | 5 (33.3) |
| **Timing planned (n, %)** | (n = 11) | (n = 8) | (n = 10) |
| Yes | 8 (73) | 8 (100) | 9 (90) |
| No | 3 (27) | | 1 (10) |
| **Medications (n, %)** | | | |
| None prescribed | 20 (80) | 11 (91.7) | 9 (60) |
| Physical health | 5 (20) | 1 (8.3) | 6 (40) |
| Mental health | | | |
| **Stressful life events (LTE total score)** | (n = 25) | | |
| Median (IQR) | 22 (20–23) | 22.5 (22–24) | 22 (22–24) |

ANC: Antenatal care, EUC: Enhanced usual care, IPV: Intimate partner violence, IQR: Inter-quartile range, LTE: List of threatening experiences (possible scores: 12 (most stressful life events) – 24 (fewest stressful life events), PST: Problem-solving therapy.

(n = 5) were moving away (n = 3), changing their mind (n = 2), or not attending appointments (n = 1). Eight out of 12 participants (67%) completed all four sessions of standard PST. Reasons for attending two (n = 1) or no sessions (n = 3) were moving away (n = 2) or not attending appointments (n = 2).

Once a fee had been agreed with ANC providers, training, supervision, and delivery of intervention sessions were acceptable; no staff dropped out of the training course or the study. The commonest reasons for ANC providers not taking part in the trial were being pregnant and expecting to move away from the study area. Some ANC providers reported that they or a pregnant woman felt uncomfortable with audio-recording sessions.

In terms of COVID-19, despite the government directive for masks, regular hand washing, and physical distancing, attending four sessions remained acceptable to pregnant women, ANC providers, and research staff.

**Appropriateness.**   The appropriateness of PST-IPV and our study design to the context was supported by mean recruitment of 10 eligible participants per week across two sites. Table 2 shows that the mean duration of each session ranged from 32 minutes (session 4) to 51

**Table 2.  Mean session duration.**

| Intervention → <br> Session number ↓ | PST-IPV <br> (minutes: seconds) | Standard PST <br> (women reporting physical and/or sexual IPV) | Standard PST <br> (women reporting psychological or emotional IPV, only) |
|---|---|---|---|
| 1 | 50:37 (n = 20) | 44:35 (n = 4) | 32:45 (n = 4) |
| 2 | 51:22 (n = 19) | 45:09 (n = 4) | 35:59 (n = 4) |
| 3 | 45:48 (n = 19) | 47:50 (n = 4) | 34:08 (n = 3) |
| 4 | 31:50 (n = 19) | 41:17 (n = 4) | 42:45 (n = 3) |

IPV: Intimate partner violence, PST: Problem-solving therapy, PST-IPV: Problem-solving therapy for women experiencing IPV.

**Table 3. Independent ratings of fidelity and therapist competence.**

| Intervention, time point | Median duration, IQR (minutes: seconds), | Median adherence score, IQR (%) | Median quality score, IQR (%) | Median responsiveness score, IQR (%) | Median ENACT score (%) |
|---|---|---|---|---|---|
| Standard PST: Accelerated case, Session 1 (n = 2) | 47:15 (42:49–51:40) | 90 (90–90) | 100 (100–100) | 100 (100–100) | 85 (76–80) |
| PST-IPV: Accelerated case, Session 1 (n = 6) | 47:19 (27:08–52:14) | 90 (86–98) | 92 (83–100) | 100 (100–100) | 76 (69–81) |
| Standard PST: random case, Session 2 (n = 3) | 45:12 (42:14–46:16) | 100 (94–100) | 100 (100–100) | 100 (100–100) | 79 (78–82) |
| PST-IPV: random case, Session 2 (n = 5) | 53:19 (43:03–53:55) | 89 (89–94) | 92 (83–100) | 75 (75–100) | 73 (73–79) |
| Standard PST: random case, Session 3 (n = 2) | 38:07 (36:38–39:36) | 86 (85–88) | 100 (100–100) | 100 (100–100) | 81 (78–83) |
| PST-IPV: random case, Session 3 (n = 6) | 47:55 (41:05–52:08) | 86 (67–89) | 100 (100–100) | 100 (100–100) | 84 (77–86) |

ENACT: ENhancing Assessment of Common Therapeutic factors rating scale, IPV: Intimate partner violence, IQR: Interquartile range, PST: Problem-solving therapy.
Minimum possible ENACT score: 33% (as 'needs improvement' is scored as 1 out of a maximum 3).

minutes (session 2). This finding raised the possibility that the content which ANC providers were instructed to cover within each session was excessive and so the intervention model may not have been appropriate for integration into ANC.

Table 2 also shows that the mean durations of the first two PST-IPV sessions were longer than those of standard PST, and that women reporting physical and/or sexual IPV randomised to standard PST appeared to have longer early sessions than women reporting psychological/emotional IPV.

**Fidelity.** Table 3 summarises fidelity and competence ratings of sessions sampled. Asking about harms and risks, assessing community or social networks, and discussing confidentiality were rated as needing improvement.

During their first clinical case, supervisors noted that ANC providers often spoke at length and offered women unsolicited advice, rather than following the manual. Supervisors acknowledged that this is common in Ethiopian ANC and reassured health workers that pausing to listen to women did not undermine their competence. One provider reported waiting until audio recording had stopped before giving advice. We organised top-up training on active listening, counselling skills, showing empathy, and key intervention components in response to supervisor feedback.

Independent fidelity assessors and supervisors noted that some ANC providers omitted intervention content if participants said they had 'no problems'. This usually occurred when initial explanations and inquiries had been brief. Supervisors noted that providers' use of the in-session flipchart visual aid varied from rigid adherence to non-use and that intervention content was sometimes repeated more than once. Some providers did not follow the recommended order, bringing content from later sessions into earlier ones.

IPV was mentioned in 50% of PST-IPV first sessions and 33% of second sessions reviewed. However, despite longer initial sessions of PST-IPV and standard PST delivered to women reporting physical and/or sexual IPV, independent assessors noted that IPV was infrequently discussed during sessions of both interventions, and when it was, it was only discussed for one to four minutes. PST-IPV supervisors noted several instances during audio-recorded sessions, in which women mentioned IPV, but health workers did not respond. One health worker expressed concern that if they discussed the traumatic events which women raised during sessions, they could become 'derailed' from planned content.

**Safety.**    Among the 29 participants (56%) who had given birth at the end of data collection, two (7%) complications (both obstructed or prolonged labour) were recorded. A single serious adverse event occurred in which a participant reported assault by her husband.

## Feasibility of measurement of outcomes and hypothesised mediators

Table 4 presents baseline and outcome assessment scores, showing both those participants randomised as intended (n = 35) and the full sample (n = 52). No concerns were raised regarding instrument comprehension and no data collectors reported undue burden on participants. Use of these measures was therefore feasible to implement.

## Discussion

As hypothesised, we identified the feasibility and acceptability of brief problem-solving therapy tailored for women experiencing IPV and of a randomised, controlled study design within rural Ethiopian ANC. However, we identified improvements to optimise the trial design to improve appropriateness, competence, and fidelity.

We report one of few feasibility studies of a brief psychological intervention for perinatal depressive symptoms in women experiencing IPV in LMICs. Unlike previously reported 10 to 12 session interventions for the mental health of women experiencing IPV in middle-income countries [20,21], PST-IPV is a four session, task-shared intervention adapted for a rural, low-income country context. Building on studies of single antenatal sessions for pregnant women experiencing common mental disorders and IPV [22], we demonstrated the feasibility of building trusting therapeutic relationships through recurring clinical encounters, integrated into ANC.

A key question for future research is whether IPV exposure should be an inclusion criterion or a secondary outcome in a fully-powered RCT, in settings of known high prevalence. In our study, 52 out of 86 (61%) women screening positive for depressive symptoms and functional impact reported experiencing IPV. Given known barriers to disclosure [72], a proportion of the remaining 34 women may also have experienced IPV. A future RCT could therefore measure IPV exposure without linking it to eligibility, similar to a Kenyan study in a region of high gender-based violence prevalence [73]. This approach would ensure that women experiencing but not disclosing IPV could still participate. In a fully-powered RCT, planned sub-group analyses could compare the efficacy of the PST-IPV model among women reporting and not reporting IPV. An American study found that avoidant coping mediated the association between negative social reactions to IPV disclosure and more severe PTSD symptoms [74]. A future RCT should therefore measure coping style as a potential confounder for any differences between the efficacy of problem-solving therapy in participants who do and do not disclose IPV exposure.

Most women completed all four intervention sessions. The acceptability of PST-IPV may have arisen from the very limited opportunities women in this context have to be listened to, encouraged, and emotionally supported: the sixth international standard of maternal and newborn healthcare [75]. Uncompassionate and disrespectful maternity care has been reported in LMICs including Ethiopia, especially towards women in rural areas [76]. Caring, respectful and compassionate (CRC) care has been prioritised [77], with a national transformation agenda for a motivated, competent and compassionate health workforce [78]. In our study, mental health specialists supervising ANC providers recommended supplementary communication skills training but independent ratings of session quality and therapist responsiveness were high, supporting the feasibility of training non-specialist staff to deliver a simplified psychological intervention in this context.

**Table 4. Descriptive scores of outcomes at baseline and follow-up: participants randomised as intended (n = 35) and all participants (n = 52).**

| Clinical score (possible score range) | Randomised as intended (n = 35) – Participants reporting physical and/or sexual IPV, with or without psychological or emotional IPV | | | Full sample (n = 52) – including 17 participants reporting psychological/emotional IPV only, who were misclassified as not reporting IPV and so not randomised to PST-IPV | | |
|---|---|---|---|---|---|---|
| | PST-IPV (Baseline (BL) n = 25, Outcome (OA) n = 19) | Standard PST (BL n = 5, OA n = 4) | EUC (BL n = 5, OA n = 4) | PST-IPV (BL n = 25, OA n = 19) | Standard PST (BL n = 12, OA n = 8) | EUC (BL n = 15, OA n = 12) |
| **Depressive symptoms** | | | | | | |
| Median total PHQ-9 score (IQR; 0–24) | | | | | | |
| Baseline | 8 (7–11) | 7 (7–10) | 11 (9–11) | 8 (7–11) | 9.5 (7–10) | 9 (6–11) |
| Outcome Assessment | 6 (3–9) | 7 (4–8) | 5.5 (0–16) | 6 (3–9) | 6 (3.5–8) | 4 (0.5–9.5) |
| **Recovery** | | | | | | |
| ≥50% reduction in total PHQ-9 (n, %) | | | | | | |
| Yes | 6 (32) | 1 (25) | 2 (50) | 6 (32) | 3 (37.5) | 6 (50) |
| No | 13 (68) | 3 (75) | 2 (50) | 13 (68) | 5 (62.5) | 6 (50) |
| **Suicidality** | | | | | | |
| Endorsing PHQ-9 Question 9 (n, %) | | | | | | |
| Baseline | 6 (24) | 2 (40) | 4 (80) | 6 (24) | 4 (33.3) | 7 (46.7) |
| Outcome Assessment | 3 (16) | 0 (0) | 1 (25) | 3 (16) | 1 (12.5) | 1 (8.3) |
| Median MINI suicidality score (IQR) if endorsing PHQ-9 Question 9 (0-58) | | | | | | |
| Baseline | 9 (1–10), n = 6 | 3 (3–3), n = 2 | 5.5 (2.5–8), n = 4 | 9 (1–10), n = 6 | 3 (2–4.5), n = 4 | 2 (1–8), n = 7 |
| Outcome Assessment | 1 (0–11), n = 3 | N/A, n = 0 | 15 (15–15), n = 1 | 1 (0–11), n = 3 | 2 (2–2), n = 1 | 11 (7–15), n = 2 |
| **Anxiety symptoms** | | | | | | |
| Median total GAD-7 score (IQR; 7–28) | | | | | | |
| Baseline | 11 (8–12) | 10 (8–10) | 13 (13–14) | 11 (8–12) | 9 (7.5–11.5) | 8 (7–12) |
| Outcome Assessment | 9 (7–12) | 8.5 (7.5–11) | 8.5 (7–16.5) | 9 (7–12) | 9.5 (7.5–11.5) | 8 (7–11.5) |
| **Post-traumatic stress symptoms** | | | | | | |
| Median total PCL-5 score (IQR; 0–80) | | | | | | |
| Baseline | 9 (4–13) | 5 (5–6) | 9 (4–16) | 9 (4–13) | 7.5 (4.5–12) | 4 (2–8) |
| Outcome Assessment | 4 (1–8) | 4 (2–5.5) | 9 (0–34.5) | 4 (1–8) | 4 (0–6) | 4.5 (0–13) |
| **Functional impact** | | | | | | |
| Median PHQ-9 Question 10 score (IQR; 0–3) | | | | | | |
| Baseline | 1 (1–2) | 1 (1–2) | 1 (1–2) | 1 (1–2), n = 24 | 1.5 (1–2) | 1 (1–2) |
| Outcome Assessment | 1 (0–1) | 1 (0.5–1.5) | 2 (2–2) | 1 (0–1) | 1 (0.5–1.5) | 1 (1–2), n = 9 |
| Median total WHODAS 2.0 score (IQR; 14–70) | | | | | | |
| Baseline | 17 (11–25) | 8 (6–11) | 31 (8–36) | 17 (11–25) | 21 (10–25) | 17 (8–28) |
| Outcome Assessment | 17 (3–25) | 22 (22–24) | 19 (6–42) | 17 (3–25) | 22 (13–24) | 14 (4–28) |
| Median days' difficulties in past 30 (0–30) | | | | | | |
| Baseline | 15 (4–30) | 6 (5–7) | 7 (4–10) | 15 (4–30) | 8.5 (6.5–16) | 7 (4–15) |
| Outcome Assessment | 4 (0–15) | 13.5 (9.5–15) | 5 (1.5–18.5) | 4 (0–15) | 9.5 (3.5–15) | 12.5 (1.5–30) |
| **Past-year IPV** | | | | | | |
| Median total NGL question score (IQR; 0–16) | | | | | | |
| Baseline | 5 (4–8) | 4 (4–4) | 5 (3–8) | 5 (4–8) | 4 (2–5) | 3 (2–5) |
| Outcome Assessment | 1 (0–8) | 1 (0–5) | 5 (1.5–9) | 1 (0–8) | 1.5 (0–5) | 1 (0–5.5) |
| Median total WHO question score (IQR; 0–49) | | | | | | |
| Baseline | 9.5 (5–11) | 6 (4–9) | 10 (9–10) | 9.5 (5–11) | 2.5 (1–6) | 3 (1–7) |
| Outcome Assessment | 3 (2–9) | 3 (1.5–4.5) | 2.5 (1–12) | 3 (2–9) | 2.5 (1.5–4.5) | 2.5 (0.5–3.5) |

*(Continued)*

**Table 4.** (Continued)

| Clinical score (possible score range) | Randomised as intended (n = 35) – Participants reporting physical and/or sexual IPV, with or without psychological or emotional IPV | | | Full sample (n = 52) – including 17 participants reporting psychological/emotional IPV only, who were misclassified as not reporting IPV and so not randomised to PST-IPV | | |
|---|---|---|---|---|---|---|
| | PST-IPV (Baseline (BL) n = 25, Outcome (OA) n = 19) | Standard PST (BL n = 5, OA n = 4) | EUC (BL n = 5, OA n = 4) | PST-IPV (BL n = 25, OA n = 19) | Standard PST (BL n = 12, OA n = 8) | EUC (BL n = 15, OA n = 12) |
| **Attitudes towards gender roles** | | | | | | |
| Median WHO question score (IQR; 16–32 (lower scores indicate less progressive views) | | | | | | |
| Baseline | 27 (24–30) | 25 (23–25) | 26 (26–30) | 27 (24–30) | 25 (22–27) | 26 (24–30) |
| Outcome Assessment | 27.5 (26–29) | 27 (25–28) | 28 (26–29) | 27.5 (26–29) | 27 (24–28) | 31 (28–34) |
| **Self-efficacy** | | | | | | |
| Median total adapted scale score (IQR; 0–20) | | | | | | |
| Baseline | 15 (10–18) | 15 (15–18) | 5 (0–10) | 15 (10–18) | 15 (10–15) | 15 (10–15) |
| Outcome Assessment | 15 (5–17) | 15 (10–17) | 15 (15–18) | 15 (5–17) | 15 (10–19) | 16 (15–20) |
| **Mastery** | | | | | | |
| Median total adapted scale score (IQR; 0–45) | | | | | | |
| Baseline | 33 (30–34) | 30 (25-32) | 35 (24–36) | 33 (30-34) | 31 (29–37) | 34 (26-36) |
| Outcome Assessment | 31 (30–36) | 32 (28–38) | 32 (23.5–41) | 31 (30–36) | 33 (27.5–37.5) | 31.5 (28–36) |
| **Perceived social support** | | | | | | |
| Median total OSSS-3 score (IQR; 3–14) | | | | | | |
| Baseline | 10 (9–11) | 11 (11–12) | 12 (10–12) | 10 (9–11) | 12 (11–12) | 11 (10–12) |
| Outcome Assessment | 10 (8–12) | 8.5 (8–11) | 12 (9–13) | 10 (8–12) | 8.5 (8–12.5) | 11 (8–12) |
| **Obstetric complications (n, %)** | (n = 9) | (n = 4) | (n = 2) | (n = 9) | (n = 8) | (n = 12) |
| Obstructed or prolonged labour | | 1 (25) | | | 1 (12.5) | 1 (8.3) |
| None documented | 9 (100) | 3 (75) | 2 (100) | 9 (100) | 7 (87.5) | 11 (91.7) |
| **Neonatal outcomes (n, %)** | (n = 9) | (n = 4) | (n = 2) | (n = 9) | (n = 8) | (n = 12) |
| Live birth without complications | 8 (89) | 3 (75) | 1 (50) | 8 (89) | 6 (75) | 9 (75) |
| None documented | 1 (11) | 1 (25) | 1 (50) | 1 (11) | 2 (25) | 3 (25) |

BL: Baseline score, GAD-7: Generalised anxiety disorder questionnaire, IPV: Intimate partner violence, MINI: Mini neuropsychiatric inventory, NGL: Non-graphic language questions, OA: Outcome assessment score, OSSS-3: Oslo social support scale, N/A: Not applicable, PCL-5: PTSD symptom checklist for DSM-5, PHQ: Patient health questionnaire, PST: Problem-solving therapy, PST-IPV: PST for women experiencing IPV, WHO: World health organization, WHODAS: WHO disability assessment schedule.

However, we found that IPV was not the primary focus of PST-IPV sessions. Although supervisors noted some instances where ANC providers missed opportunities to explore IPV further, this may also indicate the many other competing worries and stressors for which participants sought help. Although the impact of psychological interventions on subsequent IPV exposure is often measured [79], there is very little evidence from LMICs on the extent to which victim-survivors use sessions to directly address IPV-related concerns. Future research could evaluate the content of intervention sessions to quantify time spent discussing different types of problems and clarify whether providers, participants or both actively avoided talking about IPV.

An uncontrolled study of three sessions of antenatal PST in Cape Town [80] found that non-specific elements such as non-judgemental listening, discussion, and time for oneself were key benefits; perspectives on problem-solving itself were more variable. A balance is therefore needed between fidelity to the intervention manual and flexibility, to enable sessions

to meet women's needs and address their priorities [81]. A future RCT should examine the relative contributions of specific and non-specific intervention elements to beneficial outcomes.

We excluded women deemed at significant risk of self-harm or suicide. However, there is an established relationship between IPV and suicidal ideation [82–85]. Suicidality is often associated with hopelessness [86], which is relevant to the power and control exerted over women experiencing IPV, and the powerlessness described in our formative work [72]. A recent mapping review of suicide in vulnerable populations in LMICs [87] identified the need for gender-specific interventions for women addressing abuse, violence, gender-based discrimination, and access to healthcare. Suicidal ideation should therefore be investigated as a secondary outcome measure in a future RCT, as its responsivity to change may be different to that of other clinical symptoms.

Outcome assessments were completed by 75% of participants across three trial arms. To address the challenges of perinatal follow-up, a future RCT could enrol participants earlier in pregnancy, although in this Ethiopian context, most women do not attend ANC in the first trimester [88]. Conducting an immediate outcome assessment as soon as sessions ended and a follow-up assessment three months postpartum, or six months post-enrolment (as conducted by several previous trials [89–92]), may be more appropriate to cultural practices associated with childbirth in this context. Home visits for outcome assessment could also be explored but preserving confidentiality and women's safety should remain paramount. The small group which did not complete outcome assessments appeared to be younger than the group which did, highlighting the importance of optimising intervention design and trial procedures for groups known to be especially vulnerable to IPV [93]. The fact that women who did not complete outcome assessments appeared to have higher trauma symptoms than those who did is relevant to the finding that sessions did not focus on IPV and the possibility of avoidant coping.

The implementation strategy of psychiatrists and psychologists based in Addis Ababa supervising health workers in rural areas was limited by logistical barriers to regular field site visits. Other studies conducted in LMICs have adopted apprenticeship models [94] or cascade training, in which specialist master trainers train and supervise less experienced trainers [95,96]. In Ethiopia, this approach could enable rural health workers to be supervised by trainer-supervisors based in the study site, who receive online or telephone supervision from mental health specialists based in Addis Ababa. Basing supervisors in the rural district would increase the responsiveness of observations (session audio recordings would not need to be transferred to the city), and overcome logistical barriers to regular, in-person supervision.

### Protocol deviation

One third of eligible participants who reported psychological or emotional IPV at baseline were misclassified and randomised as if they had not reported IPV. A systematic review of 82 RCTs found recruitment, randomisation, and treatment errors in 39% of studies, with a median eight errors [97]. The authors highlighted inadequate reporting and recommended transparent descriptions of errors and their handling [98].

A review of perceptions of emotional IPV [99] found that laypersons, students, survivors and perpetrators in high-income countries all considered emotional IPV less severe or blameworthy than physical or sexual IPV. However, evidence suggests that psychological and emotional IPV do impact child outcomes [9] and women's mental health [100], including in LMICs [101,102].

The complexity of measurement and misclassification in RCTs of violence interventions is rarely discussed [103]. Our observation that baseline IPV scores were higher in the PST-IPV

group than in other arms suggests some potential non-random allocation, further supporting the need for automated randomisation in a future RCT. This protocol deviation highlighted entrenched preconceptions surrounding what 'counts' as IPV, and the need for more extensive training for data collectors and field supervisors.

## Clinical outcomes

Small improvements in anxiety symptoms in the PST-IPV arm and improved PTSD symptoms in both active arms are preliminary findings of interest. A meta-analysis found that anxiety improved more among women reporting IPV in LMICs following psychological intervention, than among women not reporting IPV [32], raising its importance as a therapeutic target. It has been argued that psychological interventions' therapeutic potential is reduced by neglecting the wider needs of people experiencing IPV in LMICs [16]. Trauma-informed approaches, woman-centredness, and therapeutic alliance are central to the effectiveness of interventions for women experiencing IPV [104]. A fully powered RCT should investigate the hypothesis that PST-IPV training sensitises health workers to respond to trauma symptoms, addressing hopeless, helpless, and powerless cognitions that contribute to suicidality [105].

Reported IPV reduced across all arms, post-participation but interpretation was complicated by the time periods asked about overlapping. An RCT of the common elements treatment approach delivered by lay workers in Zambia found significant reductions in self-reported IPV exposure at one [106] and two years' follow-up [107]. Qualitative interviews [108] identified safety strategies to avoid or prevent conflict and control anger, improved trust and understanding, as mediators. A future RCT should measure hypothesised interpersonal mediators of change, as well as psychological constructs, such as self-efficacy, and ensure that follow-up questions about IPV specify a shorter time period.

## Strengths and limitations

Our trial is the first of its kind in this rural Ethiopian setting. Most psychological intervention research in LMICs comes from urban or peri-urban areas of middle-income countries and few RCTs have evaluated treatments adapted for women experiencing IPV [18,20–22]. Our study benefitted from a shared control group with a separate, nested study of standard PST [109,110]. Comparing different adaptations to the same core intervention enables relative impacts on outcomes to be explored [111].

Unlike a recent study in South Africa [37], the task-sharing model was acceptable to ANC providers in this setting, who did not feel too pressured by their existing workloads to take part. The absence of financial compensation was a key barrier to successful implementation of task-shared mental health interventions in LMICs in a systematic review [112]. We paid clinicians for their time, given concerns in the literature about 'task dumping' [113], and the preliminary nature of our intervention, which is yet to be included within ANC providers' job descriptions. A range of studies highlight differences across contexts in expectations of remuneration [90,114–116] but compelling arguments have been made that not paying female volunteers to deliver services reinforces gender inequity [115]. The long-term cost, sustainability, and impacts on healthcare delivery of paying or otherwise engaging ANC providers to deliver mental health interventions must be evaluated by a future RCT.

Misclassifying women reporting psychological IPV introduced bias and would weaken the statistical power of quantitative comparisons between arms in a definitive trial. A proportion of outcome assessments taking place early introduced heterogeneity. However, both protocol deviations provided important learning about recruiting even more experienced staff and even closer supervision of field work.

This study only reports quantitative feasibility measures. A future study will analyse qualitative interviews with pregnant women who received problem-solving therapy and ANC providers who delivered it, to explore these constructs in depth,

## Conclusions

This randomised, controlled feasibility study of brief problem-solving therapy for perinatal depressive symptoms in women experiencing IPV contributes to a very sparse literature conducted in low-income countries. PST-IPV and the study design were feasible, acceptable, and safe to deliver in rural Ethiopian ANC. We identified necessary improvements to intervention fidelity and therapist competence, and considerations to optimise a future RCT.

## Supporting information

**S1 Checklist. CONSORT Checklist of information to include when reporting a pilot or feasibility trial [117].**
(DOCX)

**S1 Fig. Extended CONSORT diagram outlining how participants experiencing physical and/or sexual IPV were randomised using a three-arm table (as intended) and participants experiencing psychological/emotional IPV only were randomised a two-arm table (in error).**
(TIF)

**S1 Table. PST-IPV session content.**
(DOCX)

## Author Contributions

**Conceptualization:** Roxanne C. Keynejad, Tesera Bitew, Katherine Sorsdahl, Bronwyn Myers, Simone Honikman, Girmay Medhin, Negussie Deyessa, Louise M. Howard, Charlotte Hanlon.

**Data curation:** Roxanne C. Keynejad, Girmay Medhin, Adiyam Mulushoa, Eshcolewyine Fekadu.

**Formal analysis:** Roxanne C. Keynejad, Girmay Medhin, Charlotte Hanlon.

**Funding acquisition:** Roxanne C. Keynejad.

**Investigation:** Roxanne C. Keynejad, Tesera Bitew, Adiyam Mulushoa, Eshcolewyine Fekadu.

**Methodology:** Roxanne C. Keynejad, Tesera Bitew, Katherine Sorsdahl, Bronwyn Myers, Simone Honikman, Girmay Medhin, Negussie Deyessa, Louise M. Howard, Charlotte Hanlon.

**Project administration:** Roxanne C. Keynejad, Tesera Bitew, Adiyam Mulushoa, Eshcolewyine Fekadu.

**Resources:** Katherine Sorsdahl, Bronwyn Myers.

**Supervision:** Katherine Sorsdahl, Bronwyn Myers, Girmay Medhin, Negussie Deyessa, Louise M. Howard, Charlotte Hanlon.

**Writing – original draft:** Roxanne C. Keynejad.

**Writing – review & editing:** Roxanne C. Keynejad, Tesera Bitew, Katherine Sorsdahl, Bron-
wyn Myers, Simone Honikman, Girmay Medhin, Negussie Deyessa, Louise M. Howard,
Charlotte Hanlon.

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
