## [Decision Letter · Decision Letter 0]

21 Jul 2023

PGPH-D-23-00877

Problem-solving therapy for pregnant women experiencing depressive symptoms and intimate partner violence: a randomised, controlled feasibility trial in rural Ethiopia

Dear Dr. Keynejad,

Thank you for submitting your manuscript to PLOS Global Public Health. After careful consideration, we feel that it has merit but does not fully meet PLOS Global Public Health’s publication criteria as it currently stands. Therefore, we invite you to submit a revised version of the manuscript that addresses the points raised during the review process.

We look forward to receiving your revised manuscript.

Kind regards,

Ahmed Waqas

Academic Editor

Journal Requirements:

Additional Editor Comments (if provided):

Reviewers' comments:

Reviewer's Responses to Questions

**Comments to the Author**

1. Does this manuscript meet PLOS Global Public Health’s publication criteria? Is the manuscript technically sound, and do the data support the conclusions? The manuscript must describe methodologically and ethically rigorous research with conclusions that are appropriately drawn based on the data presented.

Reviewer #1: Yes

Reviewer #2: Yes

2. Has the statistical analysis been performed appropriately and rigorously?

Reviewer #1: Yes

Reviewer #2: Yes

3. Have the authors made all data underlying the findings in their manuscript fully available (please refer to the Data Availability Statement at the start of the manuscript PDF file)?

Reviewer #1: Yes

Reviewer #2: No

4. Is the manuscript presented in an intelligible fashion and written in standard English?

Reviewer #1: Yes

Reviewer #2: Yes

5. Review Comments to the Author

Reviewer #1: The following comments to the author are also available in the attached MS Word doc.

Peer review of

Problem-solving therapy for pregnant women experiencing depressive symptoms and intimate partner violence: a randomised, controlled feasibility trial in rural Ethiopia

General comments

This is an excellent study and well-written manuscript. The paper will form an important contribution to the literature after very minor edits, which I outline below.

Specific feedback

Abstract

Third paragraph: Fidelity to randomisation was impeded by strong cultural norms about what constituted IPV. However, the trial was feasible (recruitment rate: 1.5 per day; 37% of women screened were eligible).

Since randomization is a critical component of an RCT, it would be more accurate to state that recruitment (not the trial) was feasible.

Likewise, it would make sense in paragraph four, regarding adjustments, to address the threat to randomization along with the other adjustments listed (e.g., training to identify psychological and emotional IPV).

Introduction

Excellent work identifying and justifying the population of focus (POF), the factors (and relationship between factors) affecting the POF’s health, the gaps in the lit, the logical intervention, and the rationale for a feasibility study.

More information is needed, however, to explain why the authors believe a “brief” psychological intervention, as opposed to a more in-depth intervention, is an advantageous choice for this POF and setting. What lit corroborates this choice? What circumstances justify this choice? While I realize this was probably the subject of prior papers, please provide something here as well (including references to those prior, formative studies).

I have only one other suggestion:

First paragraph: Incident depression was associated with intimate partner violence (IPV) during pregnancy.

I suggest including the applicable inferential stat estimate here, since OR is consistently reported above and would be informative. Also, please add citation for this sentence.

Methods

Figure 1 is excellent. It is very easy to ascertain from the figure and the methods section text that bioethical requirements, including participant safety, were given foremost attention. The flow from recruitment through randomization into study arms is clear.

The methods section provides a very informative level of detail. Missing, however, is whether the participants received compensation or other incentives for their participation.

Under “outcome measurement,” the authors state: Outcome assessments were performed by masked, independent data collectors.

It was initially unclear to me if “masked” meant the assessors were blinded to participant assignment to study arm. This was clarified later. To avoid initial confusion, I suggest modifying the text to make this clear.

Again, the methods section is comprehensive and clearly demonstrates the study’s rigor. This level of commitment to methodological detail is often missing in the lit, so I particularly appreciate the effort.

Results

The results are comprehensive and articulated will in text and tables. An impressive amount of data were collected, and it is clear that the eventual RCT will likely capture sufficient data to measure direct in indirect effects. Obviously, measuring the statistical significance of pre-post differences was not applicable here. Nonetheless, the authors provide excellent summary tables that allow the reader to see how within and between subjects tests would be executed in a fully-powered RCT. Meanwhile, the study results are important and informative in their own right, addressing gaps the authors identified in the introduction.

I have no changes to suggest here.

Discussion

Paragraph three brings up important considerations regarding IPV disclosure. I agree with most of the logic. It would be interesting to stratify by IPV disclosure. However, disclosure would seem to play a powerful role in problem solving, as lack of disclosure is associated with avoidance coping. I am not suggesting changing the text, but it does appear to introduce coping style as a potential confounder.

Overall, the discussion is excellent. I have no suggestions for changes.

Reviewer #2: Review:

This article presents the feasibility testing of an adapted mental health intervention for postnatal women experiencing IPV. The article is well written and contributes important knowledge to the field of IPV intervention research. The article could be improved with some additional detail as noted in comments and questions below.

•Introduction

oIt would be good to make the case more clearly for why an intervention needs to be adapted to IPV and why PST is appropriate / evidence-based for this purpose

•Methods

oRigorous use of ADAPT framework and following MRC guidance on developing interventions prior to testing, as well as a strong 3-arm design. Appreciate the testing of IPV-adapted vs. normal PST.

oSpecify what happened for women who screened out due to illness or need for treatment

oCan you clarify that the ‘sources of support’ provided to the EUC included mental health and GBV service providers that were broadly accessible in terms of cost, location?

oNo socioeconomic indicators were collected? Food security?

oOutcomes

Did you use any frameworks to inform your definitions of feasibility and other outcomes?

Why is recruitment rate also a measure of appropriateness, in addition to feasibility? How do you distinguish it’s role in each of these?

oTheory of change and mediators

Appreciate the use of a theory of change and measurement of mediators. Did you consider acceptability of violence?

In terms of healthcare-seeking behavior, did it include seeking non-clinical services that may also be supportive for someone experience violence, e.g. legal, housing, etc.? Specify.

oTraining

Specify that the different sets of trainers was to limit contamination, if it was (p. 11)

oRandomization – I’m a little confused about how this is described. Were all women reporting IPV randomized to IPV-PST? How is that so since eligibility included reporting IPV?

oProtocol deviations – thank you for reporting this important deviation and the likely very common misclassification of GBV

•Results

oDid you look at whether characteristics of those who did not complete follow up were the same as those who did? I know it’s a small sample so looking for significant differences wouldn’t be meaningful, but from a descriptive perspective it could be interesting to know

oAcceptability

How did you assess if participants comprehended randomization? (p. 19)

Was staff drop out the only measure of their acceptability?

oAppropriateness

How did you determine a cut-off of 30 min in terms of appropriateness of integration into ANC?

oFidelity

Very important and useful findings around the issues with fidelity, these are common and therefore important to highlight for others, including the avoidance of the topic of violence among providers, interesting though that the quality scores were still quite high in Table 3, as were responsiveness. Can you comment on that in the discussion?

oSafety

What were the mechanisms of tracking and documenting safety outcomes?

oTable 4

Why did you ask about past year IPV at follow-up when it hadn’t been a year since baseline? And what was the time period used for the WHO questions? Include that information in the measures section where it would also be useful to list some example items.

It would also be helpful to describe the range for ‘attitudes towards gender roles’ and interpretation in the table footnotes; same for other outcome measures for ease of interpretation

•Discussion

oYou raise important points in the discussion, including whether IPV should be an eligibility criteria in an area of high exposure, which I think it may not need to be and your proposal for a stratified analysis is useful.

oIn terms of discussing IPV during the counselling sessions, you suggest women may not want to talk about the IPV and rather other stressors they face. Do you however have data categorizing the types of other stressors discussed by women? I imagine from the recordings you could assess that, but also glean if it was more provider avoidance than women’s desires per se. I think some more information around that would be useful and/or useful to explore in follow-on in-depth interviews with women and providers. My hunch is it was more discomfort as in my experience, women are highly interested in exploring relationship challenges faced, which intersect with other socio-economic challenges in their lives.

6. PLOS authors have the option to publish the peer review history of their article (what does this mean?). If published, this will include your full peer review and any attached files.

**Do you want your identity to be public for this peer review?** For information about this choice, including consent withdrawal, please see our Privacy Policy.

Reviewer #1: No

Reviewer #2: No

---

## [Editor Report · Decision Letter 1]

27 Sep 2023

Problem-solving therapy for pregnant women experiencing depressive symptoms and intimate partner violence: a randomised, controlled feasibility trial in rural Ethiopia

PGPH-D-23-00877R1

Dear Dr. Keynejad,

We are pleased to inform you that your manuscript 'Problem-solving therapy for pregnant women experiencing depressive symptoms and intimate partner violence: a randomised, controlled feasibility trial in rural Ethiopia' has been provisionally accepted for publication in PLOS Global Public Health.

Best regards,

Ahmed Waqas

Academic Editor